# An ultrahigh resolution pressure sensor based on percolative metal nanoparticle arrays

Minrui Chen [1], Weifeng Luo [1], Zhongqi Xu [1], Xueping Zhang [1], Bo Xie [2], Guanghou Wang [3] & Min Han [1]

Tunneling conductance among nanoparticle arrays is extremely sensitive to the spacing of nanoparticles and might be applied to fabricate ultra-sensitive sensors. Such sensors are of paramount significance for various application, such as automotive systems and consumer electronics. Here, we represent a sensitive pressure sensor which is composed of a piezo-resistive strain transducer fabricated from closely spaced nanoparticle films deposited on a flexible membrane. Benefited from this unique quantum transport mechanism, the thermal noise of the sensor decreases significantly, providing the opportunity for our devices to serve as high-performance pressure sensors with an ultrahigh resolution as fine as about 0.5 Pa and a high sensitivity of 0.13 kPa$^{-1}$. Moreover, our sensor with such an unprecedented response capability can be operated as a barometric altimeter with an altitude resolution of about 1 m. The outstanding behaviors of our devices make nanoparticle arrays for use as actuation materials for pressure measurement.

[1] National Laboratory of Solid State Microstructures, College of Engineering and Applied Sciences, and Collaborative Innovation Centre of Advanced Microstructures, Nanjing University, 210093 Nanjing, China. [2] Institute for Advanced Materials and Hubei Key Laboratory of Pollutant Analysis & Reuse Technology, Huibei Normal University, 435002 Huangshi, China. [3] National Laboratory of Solid State Microstructures, School of Physics, and Collaborative Innovation Centre of Advanced Microstructures, Nanjing University, 210093 Nanjing, China. Correspondence and requests for materials should be addressed to B.X. (email: xiebo@hbnu.edu.cn) or to M.H. (email: sjhanmin@nju.edu.cn)

Precision pressure sensors are essential to many micro-electro-mechanical systems (MEMS) devices, with applications in various areas. Currently, the development of smart systems and wearable devices has drawn tremendous attention toward the high-resolution MEMS integrated pressure sensors working stably on atmospheric pressure[1–11]. Electromechanical pressure sensors consist of two essential components: a membrane and a transducer element, which converts the applied pressure to an electrical signal change. Most recently, a wide variety of materials and nanostructures such as two-dimensional layers[1–3], nanotubes[4–6], nanofibers[7,8], nanoparticles (NPs)[9,10], and even composited conductive rubbers[11] were focused to be used in these devices. Piezoresistive sensing is the most frequently used transduction mechanism in these pressure sensors[12], owing to advantages such as direct current input, high yield, simple structure and manufacturing process, low cost, scalable, as well as easy signal collection[13,14]. These piezoresistive sensing elements undergo a change in their internal resistance when they are stressed, which breaks the ohmic contact or forms new defects in the materials. Generally, these piezoresistive sensing elements are hard to distinguish external pressure changes lower than 100 Pa, since the piezoresistive mechanism does not work if the external pressure change is tiny[15,16]. There is a tremendous interest to develop MEMS integrated pressure sensors that allow for atmospheric applications with a very high resolution of sub-10 Pa. With such a resolution, an altitude difference of about 1 m can be distinguished by barometric measurement.

Recently, percolative NP arrays have been used as piezoresistive transducers of ultrasensitive mechanical sensors, such as strain sensors[17,18], humidity sensors[19], as well as force and mass sensors[20]. In the closely spaced NP arrays, the spacing of the adjacent NPs is so small that the electron transports between NPs are dominated by tunneling or hopping[21,22]. There are a large number of percolative paths existing in the disorder NP arrays. Since the quantum tunneling or hopping is extremely sensitive to the inter-particle spacing, the percolative paths could be broken or regenerated by a tiny change in the geometries of the NP arrays. As a result, the conductance of percolative NP arrays is sensitively related to the deformation of the substrates on which the NPs deposit[17,18,20,23]. It is reasonable to assume that such mechanism is applicable to a piezoresistive pressure sensor by fabricating percolative NP arrays on flexible membranes as transducer elements.

In this paper, we realize a new configuration of piezoresistive pressure sensor fabricated from percolation-based conductive nanostructures. Differing from current piezoresistive pressure gauges, these devices transduce the external pressure on the elastic membrane on which the NPs deposited to the change of the tunneling conductance across the NP percolating networks[24–28]. The device characterizes with an extremely high resolution of about 0.5 Pa. Working as a barometric altitude sensor, it demonstrates the ability to distinguish altitude difference of about 1 m. While the majority of the piezoresistive pressure-sensing devices today use doped silicon transducers wherein they undergo a change in their carrier mobility when they are stressed, our devices offer an alternative with potentially higher pressure resolution in terms of higher sensitivity, reduced thermal disturbance, and decreased power consumption with a larger resistance of about 10 MΩ.

## Results

### Operating principle of the NP array-based pressure sensor.
Similar to typical configurations of pressure sensors, the architecture of our pressure sensor is comprised of a strain gauge fabricated directly on the surface of the membrane and hermetically encapsulated on a vacuum or gas-filled reference cavity[3,5,6,29]. A quarter cross-sectional view of our sensor is shown in Fig. 1a. For this device, the distortion of the diaphragm is sensed by a new type of piezoresistive strain gauge, which is based on the deformation-induced change in the tunneling conductance of the closely spaced NP arrays[10]. These NPs formed a discontinuous film in a disordered manner on a highly deformable membrane such as polyethylene terephthalate (PET) with prepatterned interdigital electrodes (IDEs). These can be considered as percolation pathways that conduct electric current distinguishable from the leakage current when a fixed voltage is applied[24]. The strain-sensing mechanism of this structure comes from the deformation-dependent percolation morphology over the IDEs. By applying an external pressure, a small deformation of the PET membrane induces a change in the inter-particle spacing, enabling more or fewer conductive percolation pathways, thus leading to a change in the electron conductance, as shown in Fig. 1b.

### Fabrication of the NP array-based sensing elements.
The flexible strain-sensing element (Fig. 1c) was fabricated by depositing metal NPs with controlled filling fraction carried out using a straightforward nanocluster deposition technique (the fabrication procedure is depicted in Supplementary Note 1). The conductance of the NP film was monitored during the deposition process (Supplementary Fig. 1). A typical conductance evolution curve measured during the NP deposition process is shown in Fig. 1d. It exhibits the characteristics describable with a percolation model[30,31], considering the similarity between the coverage of the NP assembly and the particle filling fraction used in the percolation model, both of which increase with the deposition time. In our device, the electrodes cover an area ranging from several square millimeters to several ten square millimeters, resulting in a huge aspect ratio (total electrode length versus electrode separation) of the inter-electrode gaps. This morphology leads to a rapid increase of the conductance after the NP coverage reaches the percolation threshold, which is determined by the electrode separation, due to the formation of a large number of conductive percolation pathways (or say closely spaced NP chains across the electrodes). Furthermore, due to the quantum tunneling nature of electron transport, the development of the conductance during NP deposition is not only dependent on the geometric filling pattern of the NPs but also dependent on the distribution of inter-particle gaps along the conductive percolation pathway, which also changes with the increase of the deposition mass. As a result, a finely gradual change in the slope of the conductance evolution curve reported in Fig. 1d can be observed in the vicinity of the percolation threshold after a few number of conductive percolation pathways are formed, which enables a least measurable electric current. With a further increase in the deposition time, the conductance evolution curve soon reaches a rapid rising slope and the gradual change of the conductance with such rising slope can span four orders of magnitude or more, rather than displaying an uncontrollable sudden drastic rising as many classical granular conductive composites show. This nature is important for the device to achieve a sensitive and high resolution response to the change in the inter-particle gap distribution induced by various physical actions, such as pressure and strain. The slope of the conductance evolution curve can be finely adjusted by the NP deposition rate, as shown in Supplementary Fig. 2. Since a precisely controlled deposition rate can be maintained in the gas phase cluster beam deposition, a reliable and reproducible production of the devices can be assured.

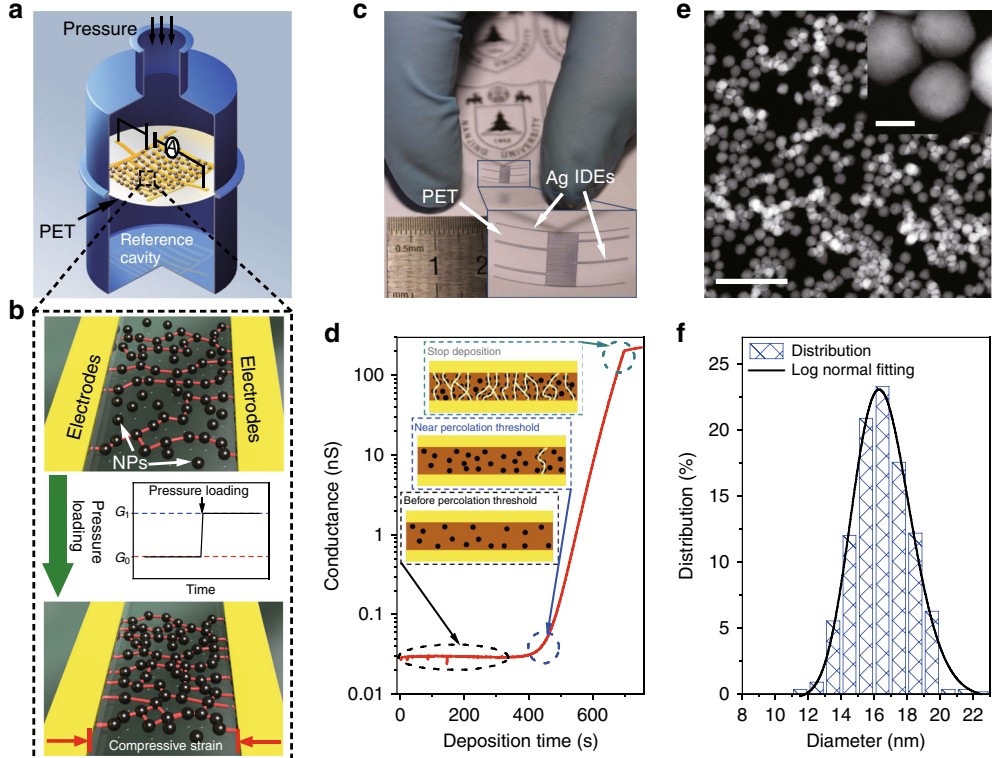

**Fig. 1** Fabrication and characterization of a pressure sensor based on the NP arrays. **a** A schematic diagram of the piezoresistive barometric pressure sensor prototype in a quarter cross-sectional view. **b** The operating principle of our sensors. These red lines linking NPs represent percolation pathways that may exist in arrays. When a pressure is loaded, more pathways are generated so that the lines become denser. **c** Photograph of the actuation layer of the sensor, consisting of a PET membrane covered with Pd NP arrays and Ag IDEs. **d** Evolution of the conductance between the IDEs during NP deposition. The insets show a schematic depiction of the inter-particle electron tunneling pathways in metal NP arrays before, close to, and beyond the percolation threshold. **e** HAADF-STEM image of the Pd NP arrays. Scale bars in image and inset represent 100 and 10 nm, respectively. **f** Size distribution of the NPs measured from **e**. The distribution fits the lognormal function. Source data are provided as a Source Data file

**Microstructural characterizations of the NP arrays**. To analyze the micro-morphological characteristics of NPs, images from the scanning transmission electron microscopy (STEM) using a high-angle annular dark field (HAADF) detector (Fig. 1e) and the scanning electron microscopy (SEM) (Supplementary Fig. 3) demonstrate uniform deposition of the NPs across the full area of the IDEs. The mean diameter of the NPs is about 16 nm (Fig. 1f). The NPs are well isolated without coalescence. Most of the NPs are closely spaced, with an inter-particle spacing smaller than 1 nm (inset image in Fig. 1e). More detail morphological information is displayed in Supplementary Note 2.

Generally, NPs of various metals can be used as the piezoresistive sensing medium. In the present research, palladium (Pd) NPs were used in preference owing to their less coalescence (Supplementary Fig. 4) and chemical stability. Although gold has been preferentially considered as a conductive and chemically stable element in many sensing applications[32,33], the high mobility and easy coalescence behavior of gold NPs leads to large instability when they were used to constitute percolative conducting NP arrays.

Oxidation of metal NPs is a common concern for a device working under atmospheric ambient directly. Oxidation of Pd NPs has been observed in a wide range of conditions[34]. For the gas phase deposited Pd NPs, our X-ray photoelectron spectroscopy (XPS) and high-resolution transmission electron microscopy (HR-TEM) characterizations demonstrated that there is a $PdO_x$ layer of 0.5 nm in thickness formed on the NP surface (Supplementary Fig. 5). Considering $PdO_x$ is a p-type semiconductor with a relatively low work function, the conductance of

the NP arrays is possible to contain the contribution from the electrons emitted from the NP surface under the applied bias voltage. However, since more than one inter-particle nanoscale gaps may be contained in most of the percolation paths and the applied bias voltage in the measurement is only 1 V, the voltage drops on each nanoscale gap should generally not be enough to stimulate the electron emission so that the conductance measured from the NP arrays is not dominated by the electron emission mechanism. Although the multi-barrier tunneling nature of electron transport in the NP array remains unchanged, the conductance of the array and therefore the response characteristic of the device will be stabilized to a new value after its exposure to air, due to the reduction of the inter-particle tunneling barrier height resulting from the formation of the $PdO_x$ surface layer with a relatively low work function.

**Response behavior of pressure sensors based on NP arrays**. A home-made system was used to test the sensing performance of the sensing elements as shown in Supplementary Note 3 and Supplementary Fig. 6. The response of the device with pressure was measured by monitoring the conductance as a function of the applied pressure, which was changed in steps. Figure 2a shows the pressure-response curves, ($\Delta G/G_0$) versus $\Delta P$, where $\Delta G = G - G_0$, in which $G$ and $G_0$ denote the conductance with and without an applied differential pressure $\Delta P$ (with reference to atmospheric pressure), respectively. We first discuss the situation of a sensor with a 0.05-mm-thick PET membrane. Over the whole applied pressure range, our sensor showed a steady response to static

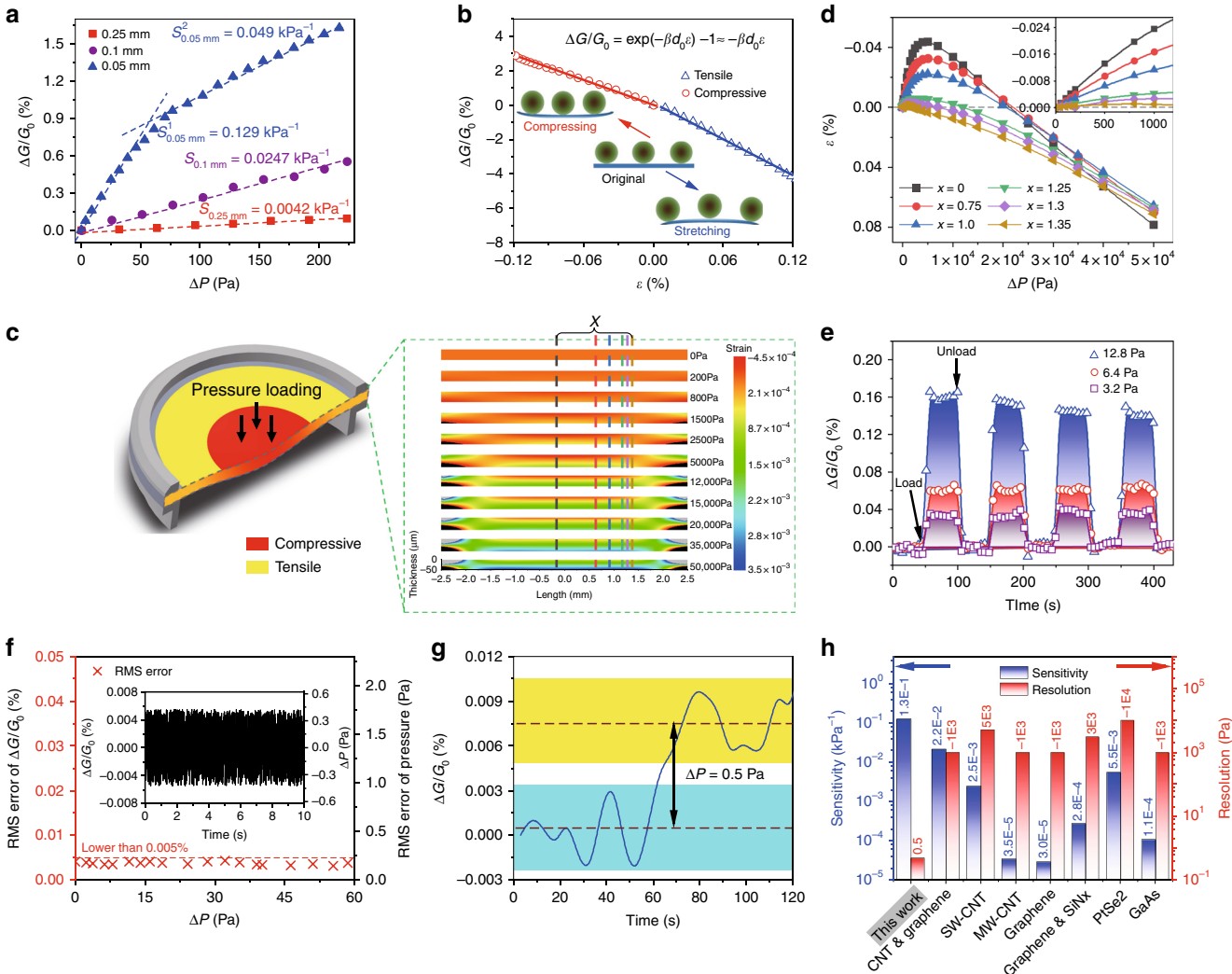

**Fig. 2** Detection of tiny differential pressures. **a** Pressure-response curves for PET membranes of different thicknesses. Error bars represent the s.d. of the conductance. **b** $\Delta G/G_0$ versus strain $\varepsilon$ curves for an actuation layer in the pressure sensor. **c** The FEA result of the strain distribution on the cross-section along the center line of the PET membrane under different pressures. **d** Mechanical deformation at various position on the upper surface of the PET membrane. $x$ denotes distance between the center of PET and the point as shown in **c**. The inset is a partial amplification of the low pressure regime. **e** $\Delta G/G_0$ response of loading and unloading cycles for different pressures. **f** The RMS error in $\Delta G/G_0$ at different $\Delta P$. The inset shows a typical $\Delta G/G_0$ fluctuation, which was used to calculate the RMS errors. Both right axes show the pressure converted from left $\Delta G/G_0$. **g** Real-time transient $\Delta G/G_0$ by applying $\Delta P = 0.5$ Pa. Data of **e**, **f**, and **g** were from measurements on a sensor with a 0.05-mm-thick PET. **h** A comparison of the sensitivity and resolution of the present device with other nanostructure-based pressure sensors. Source data are provided as a Source Data file

pressure, and the conductance under each pressure was constant (the error bars obtained from statistical analysis of the measurements were too small to be distinguished in the plot in Fig. 2a). The slope of the pressure-response curves (($\Delta G/G_0$)/$\Delta P$) could be used to characterize the sensitivity $S$ of the pressure sensor[32]. For smaller differential pressures (lower than 60 Pa), there is an approximately linear relationship between the response and the applied pressure, with a pressure sensitivity value $S = 0.13$ kPa$^{-1}$. Above 60 Pa, the sensitivity dropped to 0.049 kPa$^{-1}$. A response drop in sensitivity at higher pressures has been widely observed in recently reported pressure sensors[35–37]. In our sensor devices, the decrease in sensitivity might be attributed to the transition of the deformation behavior of the PET membrane.

For a thin film assembly of NPs on a flexible membrane, a compressive strain may induce a decrease in the mean distance to adjacent NPs, resulting in an increase in the conductance of the NP array, as shown in Fig. 1b; a tensile strain will have the opposite effect[38]. This could be proved by the changes of relative

conductance influenced by different strain, as shown in Fig. 2b. Note that the strain shown in Fig. 2b is generated homogeneously on a free PET membrane which is uniformly deformed along a single direction (the experimental details are showed in Supplementary Note 4). It can be well defined by the applied stress with an exponential correlation within the elastic limit.

Back to the pressure sensor device, the strain generated on the PET membrane under pressure is not such simple. Since the edge of the PET membrane supported on the cavity is constrained, inhomogeneous deformation is generated on the membrane under applied pressure. We find that when a pressure is applied to the PET membrane, a compressive strain is generated from the center of the membrane while a tensile strain is exerted on the surrounding area. For convenience in analysis, we pay close attention to the cross section across the center of the circular membrane, as depicted by the schematic diagram of Fig. 2c, in which the strain distribution over the cross section calculated from finite element analysis (FEA) is also demonstrated under a

series of typical pressures. The strains change with position. Their distribution on the cross section changes with the applied pressure. The strain versus pressure plot shown in Fig. 2d also changes significantly from position to position.

The inhomogeneous deformation of the membrane under pressure induces an inhomogeneous distribution of the inter-particle distance changes in the NP arrays. The conductance measured across the electrodes is an integration of the electron transport over all the conductive percolation pathways, which contain various inter-particle distances characterized with a complex function of position and pressure. Therefore, the response of conductance to pressure is no more a simple exponential function of pressure. At a smaller applied pressure, the compressive strain dominates the main area of the membrane, so that the whole NP array undergoes a conductance enhancement. An approximately linear dependence between conductance and pressure is observed. With increasing applied pressure, a transition from compressive to tensile strain can be observed as the position changes from center to edge of the membrane (see Fig. 2d and Supplementary Fig. 7). The extension of the tensile strain regime makes part of the NP array undergo a conductance reduction, which diminishes the increase in conductance generated in the central area. A drop in the conductance response curve can thus be seen. In the extreme case, tensile strain might dominate the deformation of the membrane, resulting in a decrease in the conductance with an increase in applied pressure (e.g., see the conductance response at higher pressures in Fig. 3a for a 0.05-mm-thick PET membrane). However, the decrease in sensitivity at higher pressures should not pose a significant problem for the operation of the device.

The stability and repeatability of the sensor were investigated by examining its recovery in response to repeated pressure cycles. Figure 2e shows the change in the conductance of our sensor over four loading–unloading cycles at dynamic pressures of 3.2, 6.4, and 12.8 Pa. Our sensor showed a virtually instantaneous response to increasing pressure and a highly repeatable response at each applied pressure over all the cycles. The conductance response upon pressure loading remained constant within the experimental resolution and recovered to the initial level after the pressure unloading. The amplitude of the conductance response corresponding to each pressure loading was maintained after the loading–unloading cycles. In particular, with reference to Fig. 2e, it can be seen that the relative change in conductance at 6.4 Pa applied pressure is 0.062%, with a standard deviation $\sigma$ of 0.0024%. (The four response parts of curve were collected to calculate the standard deviation.) This indicates that the sensor could resolve pressure changes as small as 1.5 Pa (within 3-sigma confidence, resolving pressure = $(6\sigma \times \Delta P)/(\Delta G/G_0)$) without difficulty. This high stability and repeatability were also demonstrated in a compression test on the NP-coated PET membrane-based strain-sensing element, wherein the conductance response characteristic did not show any evident changes after repeated compressing for over at least 500 cycles (see Supplementary Note 5 and Supplementary Fig. 8).

We now look at the resolution of the pressure measurement of the sensor. Generally, the random electrical noise in a piezo-resistive sensor, which is dominated by thermal and flicker noise, sets the fundamental lower limit of its piezoresistive transducer resolution[12]. To assess the lower limit of detection of pressure variation of the sensor, the overall sensing noise on $\Delta G/G_0$ was measured at various differential pressures. A typical fluctuation in $\Delta G/G_0$ is plotted as a function of time in the inset in Fig. 2f. When a constant pressure was maintained, the conductance fluctuated around ±0.005%, which corresponds to an uncertainty in the pressure measurement of about ±0.38 Pa (=fluctuated amplitude/S). The root mean square (RMS) noise at different applied pressures was calculated from the fluctuations in $\Delta G/G_0$, as shown in Fig. 2f[39]. It can be seen that up to 60 Pa applied differential pressure, the RMS noise is always lower than 0.005% and remains fairly constant with pressure, making the noise-limited pressure resolution of the sensor as small as 0.38 Pa. Figure 2g shows the dynamic response of the sensor exposed to an impulse of air pressure with a tiny differential pressure amplitude of 0.5 Pa. The application of the pressure impulse led to about a 0.0075% increase in G. This relative change in conductance was well above the conductance fluctuation levels, so the reversible decrease and increase in G upon loading could be clearly distinguished from the random electrical noises. This indicates that our sensor has the ability to reliably detect pressure variations as low as 0.5 Pa.

In Fig. 2h, a comparison with the state-of-the-art piezoresistive pressure sensors based on carbon nanotubes[5] (multi-wall[40] or single-wall[6,41]), graphenes[2,5,42,43], PtSe$_2$[3], and GaAs[44] is shown. It is clear that the sensitivity of our devices is among the highest category. The more remarkable is that our sensors show an excellent resolution which is nearly three orders of magnitude higher than that of most of the others[3,5,40–42,44]. It is known that the ability to detect subtle pressure variations in the regime from 1 Pa to 1 kPa is crucial for many modern applications. The ultrahigh resolution realized in this paper is a significant improvement in current sensing capabilities.

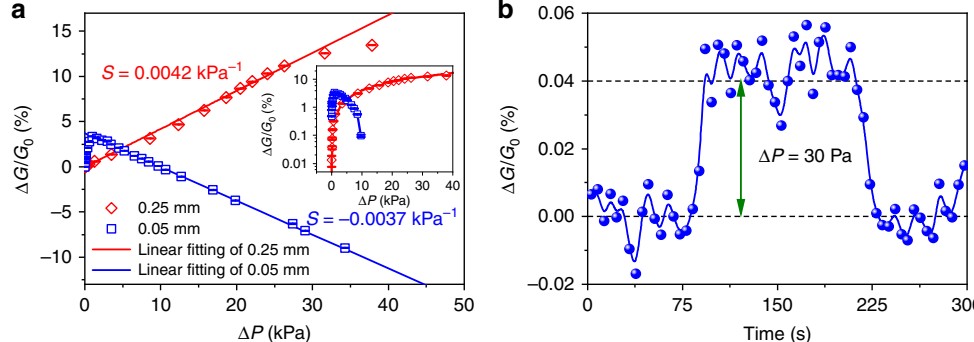

**Fig. 3** Response characteristics of sensors with actuation membranes of different thicknesses. **a** $\Delta G/G_0$ versus $\Delta P$ curves measured for the sensors having a 0.05-mm and a 0.25-mm-thick PET membrane. The inset is the logarithmic form of $\Delta G/G_0$. A pressure difference as tiny as 10 Pa could be discriminated. Error bars represent the s.d. of the conductance. **b** Real-time transient $\Delta G/G_0$ of the sensor having a 0.25-mm-thick PET membrane responding to a 30-Pa differential air pressure impulse. Source data are provided as a Source Data file

**Tailoring the response behavior with membranes**. A report has shown that modifying the mechanical and geometrical properties of the flexible substrates could change the measuring range of sensors[45]. We also investigated how the thickness of the PET membrane influenced the effective pressure regimes and sensitivity.

The thickness of the membrane is an important parameter in pressure sensors. Basically, a thinner and more robust membrane can offer higher sensitivity because it can undergo larger deformation at the same applied pressure[46]. This is true when the applied pressure is small. The pressure-response curves measured for sensors having three different PET membrane thicknesses are compared in Fig. 2a. The sensitivity is about $0.025\,\mathrm{kPa}^{-1}$ for the 0.1 mm PET membrane, and for the 0.25 mm PET membrane the sensitivity is $0.0042\,\mathrm{kPa}^{-1}$. However, at higher pressures, the sensitivity of the pressure sensor with thinner PET membrane dropped noticeably due to the expanding of the tensile strain regions, as discussed above. Since thinner films tend to be subject to significant visco-elastic creep or even plastic deformation at higher applied pressure (the plastic limit is about 1.75%, shown in Supplementary Note 6 and Supplementary Fig. 9), they have a much more limited pressure range of operation. As shown in Fig. 3a, the pressure sensor with a 0.05-mm PET membrane can work normally at 0–1.0 kPa differential pressure regime with an insignificant hysteresis (see Supplementary Fig. 10a). Beyond this pressure, the sensor could still respond well to the differential pressure up to 40.0 kPa, with a sensitivity $S = -0.0037\,\mathrm{kPa}^{-1}$ (negative sensitivity indicates that the tensile strain dominates the deformation on the PET membrane). But at higher pressure, the deformation of the membrane may be plastic so that significant hysteresis emerges

(see Supplementary Fig. 10b). On the other hand, the response curve of the sensor with a 0.25-mm PET membrane exhibits a logarithmic form over the pressure range 0–40.0 kPa shown in inset of Fig. 3a, which is in agreement with previous discussion. For this sensor, a linear response and constant sensitivity is maintained up to 30.0 kPa, beyond which the sensitivity drops significantly. This means that this sensor can operate normally with high sensitivity over a very wide pressure range. These results indicate that both sensitivity and pressure range are tunable by modifying the thickness of the PET membrane, thus the sensitivity and pressure regime requirement for different applications can be satisfied. The dynamic response of the sensor having a 0.25-mm-thick actuation layer to a pulse of pressure with a differential pressure amplitude of 30 Pa is shown in Fig. 3b. From RMS noise analysis, the sensor was found to have a lower limit of detection of pressure variation as small as 10 Pa. With that resolution, our sensors could be used as sensitive barometers.

**Application as a barometric altimeter**. To demonstrate the application of highly sensitive sensor with ultrahigh resolution in practical situations, our sensor was used to measure the altitude of a moving elevator. As is well known, atmospheric pressure varies with altitude (illustrated in Fig. 4a), and therefore, altitude can be determined based on the measurement of atmospheric pressure.

A sensor with a 0.05-mm-thick PET membrane was positioned in an elevator. It was found that during the operation of the elevator, the sensor responded to the altitude change instantaneously (Supplementary Movie 1). The greater the altitude, the lower the pressure, so the conductance clearly decreased floor by

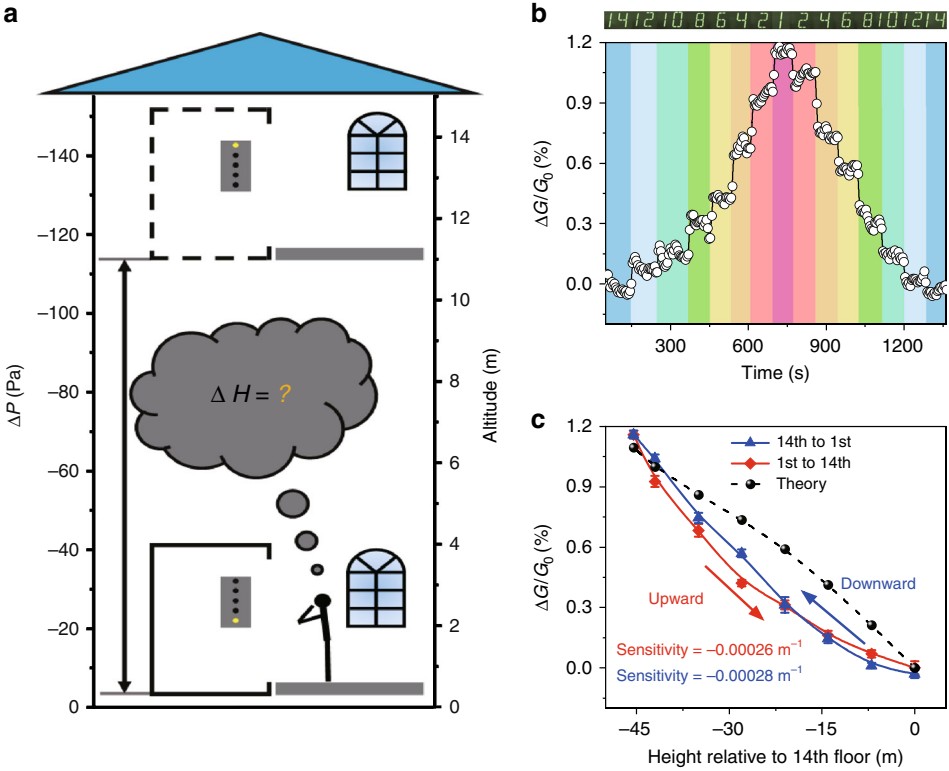

**Fig. 4** Altitude measurement with the pressure sensor. **a** A schematic diagram of differential pressure $\Delta P$ to 1 atmosphere as a function of altitude. **b** Real-time conductance changes in response to the variation in the floor elevation. The recorded signal corresponding to each floor is indicated by different colored bars. **c** Measured and calculated $\Delta G/G_0$ versus height curve. Note that $G_0$ used here is the conductance of the NP arrays measured at 14th floor. It corresponds to the conductance of the NP particle array under 500 Pa pressure. Error bars represent the s.d. of conductance. Source data are provided as a Source Data file

floor. Figure 4b shows a real-time record of our sensor response to the movement of the elevator. The elevator was started from the 14th floor, went down to the first floor and was then lifted again to the 14th floor, making a stop every two floors. The output signal of our sensor showed a virtually instantaneous response to the motion of the elevator. Stepwise conductance changes could be clearly observed. The device showed a resolution which was much higher than that was actually required to respond to a one-floor change in elevation, demonstrating that our sensor can be used as a sensitive altimeter. The relative change in the conductance plotted as a function of the change in altitude (with reference to the 14th floor) is shown in Fig. 4c. An altitude measurement sensitivity of $-0.00028\ m^{-1}$ is calculated from the slope of the response curve. An RMS noise analysis suggested that our sensor would be able to detect a change in altitude as small as 1.0 m.

Conductance response $\Delta G/G_0$ at different altitude was also calculated according to the $\Delta G/G_0$ versus pressure relationship shown in Fig. 2a and the barometric pressure calculated from a polytropic atmosphere model (see Supplementary Note 7 and Supplementary Fig. 11). As shown in Fig. 4c, there is a good agreement between the experimental $\Delta G/G_0$ and the calculated one, verifying that the altitude measurement is reliable. As a precise barometric pressure sensor, the sensor described in this paper could be applied as a high-resolution barometric altimeter; such instruments are now widely required in vehicles, aircraft, and personal navigation[47,48].

## Discussion

In summary, we have developed an efficient, low-cost approach for the fabrication of piezoresistive pressure sensors using dense Pd NP arrays deposited on flexible PET membranes, by using the hypersensitive response of the tunneling conductance of the closely spaced NP arrays to the tiny strain. Under differential barometric pressure across the diaphragm, our prototype exhibits an unprecedented noise-limited pressure resolution, which allows it to detect subtle variations in pressure of <0.5 Pa stably with an ultra-high sensitivity of 0.13 kPa$^{-1}$. Both the pressure range and sensitivity could be tunable by modifying the thickness of the PET membrane in order to meet various applied conditions, making it possible to extend the working pressure range up to 40 kPa. In addition, our sensor could distinguish an altitude difference of about 1 m when applied as a barometric altimeter in practical situations.

The high sensitivity and ultrahigh resolution realized by our sensor can be attributed to the nature of the current transport, which is dominated by electron tunneling or hopping (see Supplementary Note 8 and Supplementary Fig. 12) through percolation pathways among closely spaced NP arrays[28,49,50]. The tunneling conductance of the NP array is very sensitive to the inter-particle spacing. It can be written as $G \propto \exp(-\beta l)$, where $l$ is the spacing of the adjacent NPs and $\beta$ is a size- and temperature-dependent electron coupling term[18,51,52]. The exponential relationship means that NP arrays respond to a tiny pressure-induced deformation of the actuation membranes with atomic-scale sensitivity. Furthermore, the electrical potential energy built between adjacent NPs due to electron charging could sufficiently reduce the random transport of electrons having energy less than $k_BT$ ($k_B$ is the Boltzmann constant), which contribute to the lower energy portion of the statistical distribution of electron energy, although no Coulomb blockade was observable at room temperature. As a result, thermal noise may decrease significantly, which enables an increased sensing resolution.

Although in this study the superior sensing capabilities were demonstrated for the measurement of barometric pressure, we believe that the sensing mode and fabrication methodologies of this new low-cost piezoresistive pressure sensor can also be used in the design of other flexible low-cost pressure sensors for application in a broad range of fields, such as wearable healthcare systems and ultra-sensitive e-skins.

## Methods

**Device fabrication**. The fabrication of sensing elements is shown in Supplementary Note 1 and Supplementary Fig. 1 in detail. PET membrane without any scar was washed with alcohol and deionized water. Silver IDEs were deposited on the membrane by shadow mask evaporation in high vacuum. The electrodes contain an electrode separation of about 15 μm, with an as-prepared resistance large than $10^{10}\ \Omega$. Electrodes covered an area ranging from several square millimeters to several ten square millimeters, resulting in a huge aspect ratio (total electrode length versus electrode separation) of the inter-electrode gaps. Pd NPs were generated from a home-made magnetron plasma gas aggregation cluster source in argon stream at a pressure of about 80 Pa and extracted to a high vacuum deposition chamber with a differential pumping system. Some deposition parameters are displayed in Supplementary Table 1. The Pd NPs were deposited on the PET membrane fabricated with IDEs. The whole area of the electrodes was uniformly covered with the deposited NPs. A constant deposition rate could be precisely maintained and monitored with a quartz crystal microbalance. During the deposition, the electric current across the IDE gaps was measured in real-time under 1.0 V applied bias with a source meter (Keithley 2400). A typical plot of the conductance evolution of the NP arrays as a function of deposition time is shown in Fig. 1d. The deposition was stopped when the predetermined conductance values were attained. The piezoresistive pressure sensor was completed by mounting the PET membrane on a reference cavity using Teflon O-rings.

**Response behavior of the pressure sensor**. The piezoresistive pressure sensor was connected to a pressure controller for applying different pressures (see Supplementary Note 3 and Supplementary Fig. 6) on the sensors. The electric conductance of the NP arrays on the actuation membrane was measured at different pressures with a source meter (Keithley 2400) connected electrically to the IDEs through vacuum-compatible feedthroughs that were drawn outside the cavity. A specially designed setup containing a larger volume stainless-steel chamber connected with thin (8 mm diameter) bellows was used as the precise pressure controller. The volume of the larger chamber could be chosen from 6 mL to 7.8 L. The length of the bellows could be finely adjusted with a long travel micrometer drive to generate small volume changes in the whole setup. Under static state condition, the small volume change induced a small pressure change, which could be calculated simply by employing the ideal gas model assumption. When the chamber was pre-filled with 1 atm air, such a system could generate a static pressure change around the atmospheric pressure with an extremely high resolution of 0.1 Pa. The hysteresis in sensors could be measured with this home-made system (see Supplementary Note 9).

**FEA simulation of the strain on the membrane**. The commercial software ANSYS 19 was chosen to perform FEA (see Supplementary Note 10). The flexible PET substrate used in the pressure sensor was modeled as a disk with actual geometrical dimensions of 50 μm thickness and 5 mm diameter. A Young's modulus of 2.8 GPa and a Poisson ratio of 0.38 were assigned to the PET. The edge of the disk was fixed.

**Measurement of conductance versus strain**. The actuation layer was removed from the sensor. Strains were generated from the deformations of the actuation layer which was subjected to a micrometer step by step (see Supplementary Fig. 13). When the micrometer worked on the surface that the NPs deposited, the compressive strain was generated while a tensile strain was generated by working on the back surface. The quantity of ε could be calculated (more details are given in Supplementary Note 4).

## Data availability

The data that support the findings of this study are available from the corresponding author upon request.

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

## Acknowledgements

The authors thank the National Natural Science Foundation of China (Grant nos. 11627806 and 51871091), the National Key R&D Program of China (Grant no. 2016YFA0201002), and the Hubei Key Laboratory of Pollutant Analysis and Reuse Technique (Hubei Normal University) (Grant no. PA20170204). This research was also supported by a project funded by the Priority Academic Programme Development of Jiangsu Higher Education Institutions.

## Author contributions

M.C. planned and carried out all the experiments, analyzed the data, and wrote the manuscript partly. X.Z. executed the STEM and SEM characterization. W.L. helped with experiments of the response characteristic of the sensors. Z.X. provided support for the experiment of altitude measuring in elevator. B.X. conceived the experiments and was involved in all the research phases. G.W. participated in discussing the data and editing the manuscript. M.H. conceived and supervised the project, collected and analyzed the data, and wrote the manuscript mainly. All the authors contributed to and commented on this paper.

## Additional information

**Competing interests:** The authors declare no competing interests.

**Peer Review Information**: *Nature Communications* thanks the anonymous reviewers for their contribution to the peer review of this work. Peer reviewer reports are available.

