## [Peer Review File · Nature Communications]

Reviewers' comments:

Reviewer #2 (Remarks to the Author):

The manuscript titled, "An ultrahigh resolution pressure sensor based on percolative metal nanoparticle array" by Chen et al. describes a diaphragm with a nanoparticle array to sense the deformation that is correlated to the applied pressure. The high sensitivity to measure deformation (due to pressure) is attributed to the multiple tunnel junctions in the array of nanoparticles that depend exponentially on interparticle distance. The multi-tunnel junction characteristics of the metallic nanoparticle array are evidenced by the increase in thermal conductivity with temperature (Fig. S7).

Overall, the study is carefully performed, and the phenomenological results are interesting (Fig. 4). However, the physics and the mechanics of the study require major explanations:

1. The physics of relating the strain to conductance is puzzling. The author explain the high sensitivity by claiming that owing to multiple electron tunneling junctions in the along the percolation path in the array, the conductance is exponentially sensitive to average interparticle distance, d_0 (Eq. S5 in SI). Thus, presumably, as the array deforms (due to pressure), d_0 changes leading to large change in conductance. If the deformation is affine (i.e., no slippage between the particles and the array, a valid assumption considering low hysteresis noted in Fig. 2(c)), the conductance should rise rapidly and non-linearly with pressure. However, the observation is linear dependence between pressure and conductance change (Fig. 2(a) and Fig. 2(b)). Why?

2. It appears from the mechanics of the polymer membrane (Fig. S6 in SI) that the deformation become inelastic beyond strain of $\sim 1.5 \times 10^2$. Fig. 2(b) shows, smooth, linear behavior up to a strain of 1×10^3 which is thru the the elastic-plastic (non reversible) regime. This requires some explanation.

3. Along the same lines as above, Fig. 2(a) has a transition at $\Delta G/G \sim 0.9$; while in Fig. 2(b) the curve goes smoothly and linearly up to $\Delta G/G$ of ± 3 . I realize the x-axes are different in the two figures, but if we assume the deformation of the array is affine (i.e., commensurate with deformation of the polymer) I would have expected some abrupt change around $\Delta G/G \sim 0.9$ in Fig. 2(b). Some explanation would be helpful.

4. I am a bit confused about compressive and tensile deformation occurring in different parts of the membrane. It appears the authors are trying to assert that in the center the array is compressed (i.e., G increases) while off-center array extend (lines 137-138). A schematic explaining the non-uniformity would be helpful.

5. A curiosity. If the building is well air conditioned, i.e., temperature is isothermal, the pressure will change exponentially with height. Their Fig. 4 does exhibit some non-linearity. Can the authors, from an exponential fit back out the value of g (if T is known)?

6. A curiosity. How does the hysteresis in Fig. 2(c) look for higher pressure? For example close to, and beyond, 50 KPa where the deformation may become plastic?

7. It seems the actual invention is the device with array on PET polymer. For example see, Segev-Bar, Tunable Touch Sensor and Combined Sensing Platform: Toward Nanoparticle-based Electronic Skin, DOI: 10.1021/am400757q. Is there any special reason for good particle/polymer adhesion? Why not use Au nanoparticles?

8. Incomplete literature. The idea of modulating the conductance of nanoparticle array by deforming the underlining membrane is shown before to sensitively measure humidity and pressure variation due to sound (see Berry et al., *Angewandte Chemi Int. Ed.* 2005, 44, 6668;

Nano Letters 2004, 4, 939).

9. Minor issues: (a) Personally, I find the description "quantum conductance" somewhat odd terminology. (Generally, as the authors well knows, conductance is quantum mechanical). May be think of a better descriptor. (b) The term "pressure-dependent deformation" on line 132 does not mean anything to me. A more detailed explanation of the abrupt drop in sensitivity and yet maintaining linearity in Fig. 2(a) would be helpful.

In my opinion the device is clever and the experimental study is good. More thinking and better explanation is required to underscore the underlining physics of the device to explain the high sensitivity and the measured responses.

Reviewer #3 (Remarks to the Author):

This manuscript reports about the use of a nanoparticle array to fabricate an ultra sensitive pressure sensor. Piezoresistive properties of the array are exploited. The use of arrays of nano objects near the electrical percolation threshold to detect pressure variation or gas traces has been proposed by many authors, in the majority of cases using nano composites.

The authors have already published the proof-of-principle of their approach based on the use of layers made by nanoparticles produced in the gas phase in ref. 17 of this manuscript. Here they describe the characterisation of a device showing very interesting performances compared to the state of the art of similar devices.

The authors claim that quantum conductance is the mechanism responsible for the extremely high sensitivity of their device, however the physical aspects of this claim are not discussed. In particular the use of Pd particles should be taken into account from this point of view since Pd nanoparticles are prone to oxidation in a very wide range of conditions (J. Phys. Chem. C 2007, 111, 2, 938-949). Considering the production conditions used by the authors, one should expect to have oxidised Pd nanoparticles and PdOx is a p-type semiconductor with a relatively low work function. These aspects should be discussed in order to provide convincing motivations about the conduction mechanisms.

Another very important aspect is the percolation curve reported in the supplementary information section (Figure S2). The shape of this curve should depend on the nanoparticle deposition conditions: the control and reproducibility of these parameters and of the percolation curve should be discussed in the main text of the manuscript, since this is a very important aspect related to the possibility of producing reliable and reproducible devices. A critical issue is the change of slope of the curve reported in Fig. S2 from which the authors determine when to stop the film growth. The authors should discuss in detail the physical reasons of this sudden change in slope since these are connected with the nature of the electrical behaviour of the sensor and with the mechanism (quantum conductance?).

Without a careful discussion of these aspects, the manuscripts lacks of new significant physical insights compared to what already published by the authors.

Reviewer #2

Allow me to express to you the authors' sincere thanks for your careful reviews and good comments.

1. The physics of relating the strain to conductance is puzzling. The author explain the high sensitivity by claiming that owing to multiple electron tunneling junctions in the along the percolation path in the array, the conductance is exponentially sensitive to average interparticle distance, do (Eq. S5 in SI). Thus, presumably, as the array deforms (due to pressure), do changes leading to large change in conductance. If the deformation is affine (i.e., no slippage between the particles and the array, a valid assumption considering low hysteresis noted in Fig. 2(c)), the conductance should rise rapidly and non-linearly with pressure. However, the observation is linear dependence between pressure and conductance change (Fig. 2(a) and Fig. 2(b)). Why?

Answer: Since the electrodes used in our device have a huge aspect ratio (total electrode length vs. electrode separation), the measured conductance is contributed by many conductive percolation pathways, each of which contains multiple electron tunneling junctions characterized with certain inter-particle distances. If the PET substrate is uniformly deformed, all the inter-particle distances contained in the NP array will have same relative changes. In this case, the conductance is indeed exponentially sensitive to the average inter-particle distance, which changes proportionally with the strain generated on the PET surface. According to a simple physical model (Appl. Phys. Lett. 91, 183105 (2007) and ACS Nano 2011, 5, 8, 6516-6526), the relationship between the relative change of the conductance and the strain can be written as $\Delta G/G_0 \approx \exp(-\beta d_0 \varepsilon_{\text{strain}}) - 1$, where d_0 is the average inter-particle distance and β is a size- and temperature- dependent electron coupling term. At small strains, such as $\varepsilon < 0.08\%$, a linear approximation can be applied, which gives $\Delta G/G_0 \approx -\beta d_0 \varepsilon_{\text{strain}}$. As shown in Fig. 2d in the revised manuscript, the $\Delta G/G_0$ versus strain curve measured from a free PET membrane which is uniformly deformed along a single direction (also refer to the response to comment 3 for detail) can be well fitted with the above function. At small strains,

the curve looks actually quite linear, either under tensile deformation or under compressive deformation.

However, in the pressure sensor device, the edge of the PET membrane is constrained, resulting in inhomogeneous deformation of the membrane under applied pressure. As can be visualized with a finite element analysis (refer to Supplementary Section 2.1 and the response to comment 4 for detail), there is an isotropic distribution of strains on the PET membrane plane. In Fig. 2b, the inhomogeneous distributions of the surface strain at different applied pressure are shown. Not only the strains change with position, but also their distribution changes with pressure, especially at higher pressure, in which case a transition from compressive to tensile strain can be observed as the position changes from center to edge of the membrane. As shown in Fig. 2c, the strain versus pressure plot also changes significantly from position to position. Since the deformation is affine, the inhomogeneous deformation of the membrane induces an inhomogeneous distribution of the inter-particle distance changes in the NP arrays under a certain pressure. Considering that the conductance measured across the electrodes is an integration of the electron transport over all the conductive percolation pathways, which contain various inter-particle distances characterized with a complex function of position and pressure, it is not necessary to assume an exponential or non-linear correlation between the conductance change and pressure. In fact, an approximately linear dependence between conductance and pressure can only be observed in a small pressure regime. At higher pressure, the $\Delta G/G_0$ versus pressure curve changes its slope significantly.

Fig. 2 has been modified:

Figure Fig. 2. Detection of tiny differential pressures. (a) Pressure–response curves ($\Delta G/G_0$ versus ΔP) for PET membranes of thickness 0.05, 0.1, and 0.25 mm. Dashed lines serve to guide the eye. (b) The FEA result of the strain distribution on the cross-section along the center line of the PET membrane under different pressures. (c) Mechanical deformation at various position on the upper surface of the PET membrane. The distances (x 's) between the central of PET and points as shown in (b) are 0, 0.75, 1.0, 1.25, 1.3 and 1.35 mm, respectively. The inset is a partial amplification in the pressure regime from 0 to 1000 Pa. (d) $\Delta G/G_0$ versus strain ϵ curves for an actuation layer in the pressure sensor. The solid Dashed lines serve to guide the eye and present linear dependence of $\Delta G/G_0$ on absolute ϵ . (ee) Plots of $\Delta G/G_0$ response as a function of time after loading and unloading for three applied pressures (3.2, 6.4, and 12.8 Pa). (df) The RMS error in $\Delta G/G_0$ at different ΔP ; the right-hand axis shows the applied pressure profile calculated from $\Delta G/G_0$; the inset shows a typical $\Delta G/G_0$ fluctuation plotted as a function of time, which was used to calculate the RMS errors. (eg) Real-time transient $\Delta G/G_0$ by applying a tiny pressure ($\Delta P = 0.5$ Pa). Datum from (ee, df and eg) were tested by using a sensor with a PET membrane thickness of 0.05 mm. (fh) A comparison of the sensitivity and resolution of the present device with other nanostructure based pressure sensors. The sign \sim before the value indicates that it was approximatively calculated from the datum in the references. In terms of the resolution, the smaller the value, the performance is better.

Accordingly, we have revised the corresponding paragraph of the manuscript as blow (line 179, page 8):

...For smaller differential pressures (< 60 Pa), there is an approximately linear relationship between the response and the applied pressure, with a pressure sensitivity value $S = 0.13 \text{ kPa}^{-1}$. Above 60 Pa, the sensitivity dropped to 0.049 kPa^{-1} . A response drop in sensitivity at higher pressures has been widely observed in recently reported pressure sensors.³⁷⁻³⁹ In our sensor devices, the decrease in sensitivity might be attributed to the transition of the pressure-dependent deformation behavior of the PET membrane.

For a thin film assembly of NPs on a flexible membrane, a compressive strain may induce a decrease in the mean distance to adjacent NPs, resulting in an increase in the conductance of the NP array, as shown in Figure-Fig. 1e1b; a tensile strain will have the opposite effect.⁴⁰ This could be proved by the changes of relative conductance influenced by different strain, as shown in Figure-Fig. 2b2d. Note that the strain shown in Fig. 2d is generated homogeneously on a free PET membrane which is uniformly deformed along a single direction (the experimental details are showed in Supplementary Section 1.4). It can be well defined by the applied stress with an exponential correlation within the elastic limit.

Back to the pressure sensor device, the strain generated on the PET membrane under pressure is not such simple. Since the edge of the PET membrane supported on the cavity is constrained, inhomogeneous deformation is generated on the membrane under applied pressure. We find that ~~When~~ when a pressure is applied to the PET membrane ~~supported on the cavity~~, a compressive strain is generated from the center of the membrane while a tensile strain is exerted on the surrounding area. For convenience in analysis, we pay close attention to the cross section across the center of the circular membrane, as depicted by the schematic diagram of Fig. 2b, in which the strain distribution over the cross section calculated from Finite Element Analysis (FEA) is also demonstrated under a series of typical pressures. The strains change with position. Their distribution on the cross-section changes with the applied pressure. The strain versus pressure plot shown in Fig. 2c also changes significantly from position to position.

The inhomogeneous deformation of the membrane under pressure induces an inhomogeneous distribution of the inter-particle distance changes in the NP arrays. The conductance measured across the electrodes is an integration of the electron transport over all the conductive percolation pathways which contain various inter-particle distances characterized with a complex function of position and pressure. Therefore, the response of conductance to pressure is no more a simple exponential function of pressure. At a smaller applied pressure, the compressive strain dominates the main area of the membrane, so that the whole NP array undergoes a conductance enhancement. An approximately linear dependence between conductance and pressure is observed. With increasing applied pressure, a transition from compressive to tensile strain can be observed as the position changes from center to edge of the membrane (see Fig. 2c). The extension of the tensile strain regime makes part of the NP array undergo a conductance reduction, which diminishes the increase in conductance generated in the central area. A drop on the conductance response curve can thus be seen. the area of tensile strain extends toward the center of the membrane, inducing part of the NP array undergoing a conductance reduction, which diminishes the increase in conductance generated in the central area. In the extreme case, tensile strain might dominate the deformation of the membrane, resulting in a decrease in the conductance with an increase in applied pressure (e.g., see the conductance response at higher pressures in ~~Figure Fig.~~ 3a for a 0.05 mm thick PET membrane). However, the decrease in sensitivity at higher pressures should not pose a significant problem for the operation of the device.

2. It appears from the mechanics of the polymer membrane (Fig. S6 in SI) that the deformation become inelastic beyond strain of $\sim 1.5 \times 10^2$. Fig. 2(b) shows, smooth, linear behavior up to a strain of 1×10^3 which is thru the elastic-plastic (non reversible) regime. This requires some explanation.

Answer: The reviewer's confusion may come from the inappropriate drawing format we used. The negative sign in the axes title may not be notable enough. In fact, the strain in Fig. 2b in the original manuscript varies from 0 to $\pm 1.2 \times 10^{-3}$, which is much smaller than the yield strength ($\sim 1.75 \times 10^{-2}$ as can be read from Fig. S6 of the original Supplementary) of the PET

membrane. To make a clear presentation, both Fig. 2b and Fig. S6 have been revised. In the revised manuscript, they become Fig. 2d and Fig. S10 respectively.

3. Along the same lines as above, Fig. 2(a) has a transition at $\Delta G/G \sim 0.9$; while in Fig. 2(b) the curve goes smoothly and linearly up to $\Delta G/G$ of ± 3 . I realize the x-axes are different in the two fig.s, but if we assume the deformation of the array is affine (i.e., commensurate with deformation of the polymer) I would have expected some abrupt change around $\Delta G/G \sim 0.9$ in Fig. 2(b). Some explanation would be helpful.

Answer: The authors apologize for the absence of the experimental details about Fig. 2b in the original manuscript. In fact, Fig. 2a and Fig. 2b in the original manuscript were obtained at different experimental configurations. Fig. 2b was used to illustrate the response characteristics of the conductance of the NP arrays to tensile or compressive strains. For this purpose, a conductance versus strain curve measured with the real configuration of the pressure device was unsuitable since it was difficult and inaccurate to determine a strain which was inhomogeneously distributed on the surface, as we noted in the response of comment 1. Instead, Fig. 2b was measured from a free PET membrane which was uniformly deformed along a single direction under the applied stress on a home-made apparatus. The detail of the measurement is shown in Supplementary Section 1.4. In this case, the strain was homogeneously distributed and well defined. It changed linearly with the applied stress within the elastic limit (see Fig. S10). As a result, an exponential correlation between the conductance change and the stress could be expected. On the other hand, Fig 2a was measured with a real configuration of the sensor device. The edge of the PET membrane was strictly constrained, resulting in an inhomogeneous deformation of the membrane under applied pressure. According to the finite element analysis (refer to Supplementary Section 2.1 and the response to comment 4 for detail), the strain generated on the membrane surface was a complex function of the position and pressure (see Fig. 2b and 2c). Furthermore, a transition from compressive to tensile strain occurred somewhere on the membrane surface with the increase of the pressure. This induced an inhomogeneous distribution of the inter-particle distance changes in the NP arrays under a certain pressure. Since the conductance measured across the electrodes was an integration of

the electron transport over all the conductive percolation pathways which contained various inter-particle distances characterized with a complex function of position and pressure, the conductance versus pressure curve displayed some complex features, such as the abrupt change on the slope around $\Delta G/G_0 \sim 0.9$ that can be observed in Fig. 2a. Such abrupt change might be arisen from the change of the distribution of compressive and tensile strains in accompany with the pressure increasing.

We have added the experimental details for Fig. 2b in Supplementary Section 1.4:(line 119, page 8)

The strain sensing behaviors of Pd NP films were investigated by subjecting an unconstrained PET substrate to a series of bending operations which were produced by a home-made measuring apparatus as shown in Fig. S7a. The PET membrane based actuation layer was placed above the U-type holder freely, and deformed by the micrometer with an accuracy of 1 μm . The strains induced by the deformations can be expressed as follows:^{S3}

$$\varepsilon = \frac{T_s}{2r_b} \quad (\text{Eq. S1})$$

where T_s is the substrate thickness and r_b is the radius of curvature of the membrane which is bent uniformly. Considering the geometric relationship showed in Fig. S7b, r_b can be calculated from the following equation:

$$r_b^2 = \left(\frac{l}{2}\right)^2 + (r_b - h)^2 \quad (\text{Eq. S2})$$

where the value of l can be measured easily, and the value of h can be read out from the micrometer. Meanwhile, the conductance variations of the Pd NP arrays were acquired by the source meter (Keithley 2400).

Fig. S7 Schematic diagrams of strain sensing measurements. (a) Measurement configuration of the strain sensing test. (b) Geometrical relationship used to calculate the strain. The dash line corresponds to the neutral plane of the PET membrane.

As depicted in Fig. S7, the two ends of the PET membrane could slide freely on the U-type holder. With the bending, a uniform strain was generated and induced a change on the relative conductance.

Some explains were added in the manuscript. Moreover, in order to make a more clear presentation of the relationship between the relative conductance change and strain, Fig. 2b in the original manuscript was replaced by Fig. 2d in the revised manuscript. We stated in the revised manuscript (line 187 , page 8):

...This could be proved by the changes of relative conductance influenced by different strain, as shown in Figure Fig. 2b2d. Note that the strain shown in Fig. 2d is generated homogeneously on a free PET membrane which is uniformly deformed along a single direction (the experimental details are showed in Supplementary Section 1.4). It can be well defined by the applied stress with an exponential correlation within the elastic limit.

And (line 209, page 9):

At a smaller applied pressure, the compressive strain dominates the main area of the membrane, so that the whole NP array undergoes a conductance enhancement. An approximately linear dependence between conductance and pressure is observed. With increasing applied pressure, a transition from compressive to tensile strain can be observed as the position changes from center to edge of the membrane (see Fig. 2c). The extension of the tensile strain regime makes part of the NP array undergo a conductance reduction, which diminishes the increase in conductance generated in the central area. A drop on the conductance response curve can thus be seen.

4. I am a bit confused about compressive and tensile deformation occurring in different parts of the membrane. It appears the authors are trying to assert that in the center the array is compressed (i.e., G increases) while off-center array extend (lines 137-138). A schematic explaining the non-uniformity would be helpful.

Answer: That's true. At small applied pressure, compressive deformation dominates the whole area. With the increase of the pressure, tensile deformation develops from the edge of the nanoparticle array, while the center of the array remains compressed. The off-center tensile deformation area spreads with the increase of the pressure and at large applied pressure tensile deformation dominates the whole array. In the revised manuscript, we added a schematic (Fig. 2b) to explain the strain distribution and evolution behavior. Our speculation on the non-uniform distribution and evolution of the strain is also supported by a Finite Element Analysis (FEA) modeling. Some of the FEA calculation results were used in the schematic. The detail of the FEA analysis was given in Supplementary Section 2.1: (line 141, page 9)

2.1 FEA simulation of the strain on the diaphragm

To determine the characteristics of the pressure sensor, we performed FEA using the commercial software ANSYS 19. The flexible PET substrate used in the pressure sensor was modelled as a disk with actual geometrical dimensions of 50 μm thickness and 5 mm diameter. A Young's modulus of 2.8 GPa and a Poisson ratio of 0.38 were assigned to the PET. The edge of the disk was fixed. Figure S8 shows the evolution of strain on the upper surface of the PET membrane, where the NPs deposited, under different pressures. It should be noted that the strain at 0 Pa came from the extrusion of clamp on the edge of the diaphragm.

Fig. S8 Finite element analysis modeling shows the evolution of strain on PET substrate under different pressures. It confirms that there is a transition from compressive strains to tensile strains with the increase of the pressure.

In the revised manuscript, evolutions of strains with the applied pressures were also shown (in Fig. 2c) for some typical points on the upper surface of the PET membrane along the membrane cross section. A transition from compressive to tensile strain can be clearly observed. A statement has been given in the revised manuscript (paragraph 198, page 8):

...For convenience in analysis, we pay close attention to the cross section across the center of the circular membrane, as depicted by the schematic diagram of Fig. 2b, in which the strain distribution over the cross section calculated from Finite Element Analysis (FEA) is also demonstrated under a series of typical pressures. The strains change with the position. Their

distribution on the cross-section changes with the applied pressure. The strain versus pressure plot shown in Fig. 2c also changes significantly from position to position.

5. A curiosity. If the building is well air conditioned, i.e., temperature is isothermal, the pressure will change exponentially with height. Their Fig. 4 does exhibit some non-linearity. Can the authors, from an exponential fit back out the value of g (if T is known)?

Answer: By using a polytropic atmosphere model, the pressure (relative to the pressure at 14th floor) at different height (relative to the altitude of 14th floor) was calculated and showed in Fig. S9. By applying the $\Delta G/G_0$ versus pressure relationship (Fig. 2a) measured for the sensor with a 0.05 mm thickness PET membrane, a $\Delta G/G_0$ versus height curve was calculated (refer to Supplementary Section 2.2 for detail). The result was shown in Fig. 4c, together with the measured $\Delta G/G_0$ versus height curves. Considering the unideal environment conditions, the experimental $\Delta G/G_0$ might be influenced by many uncertainties induced by temperature fluctuation, gas flow, height determination, vibration and so on, the agreement between the experimental $\Delta G/G_0$ and the calculated one was acceptable.

Fig. S9. A pressure verse height curve calculated based on a polytropic atmosphere model.

Figure-Fig. 4. Altitude measurement with the pressure sensor. (a) A schematic diagram of differential pressure ΔP to 1 atmosphere as a function of altitude. (b) Real-time conductance changes in response to the variation in the floor elevation measured by a pressure sensor positioned in an elevator. The recorded signal corresponding to each floor is indicated by different colored bars. (c) Measured and calculated $\Delta G/G_0$ versus height curve. Note that G_0 used here is the conductance of the NP arrays measured at 14th floor. It corresponds to the conductance of the NP particle array under 500 Pa pressure. Plots of the relationship between $\Delta G/G_0$ and altitude change relative to that of the 14th floor.

The calculation of the $\Delta G/G_0$ versus height curve by applying the measured $\Delta G/G_0$ versus pressure relationship was detailed in the revised Supplementary (Section 2.2, line 154, page 11):

In the barometric altimeter experiment, $\Delta G/G_0$ at each floor can also be calculated from the $\Delta G/G_0$ versus pressure relationship presented in Fig. 2a, and used to compare with the experimental ones. To do such calculation, a pressure versus height curve was firstly calculated from the polytropic atmosphere model (See: website of the National Oceanic and Atmospheric

Administration (NOAA) (<https://www.weather.gov/media/epz/wxcalc/pressureAltitude.pdf>) with the following formula,

$$P = 101325 \times \sqrt[0.190284]{1 - \frac{h}{0.3048 \times 145366.45}} \quad (\text{Eq. S3})$$

where P is the barometric pressure at altitude h .

In the calculation, a local altitude of about 25 m (data from Google Earth) was used. In Fig. S9, the calculated pressure (relative to the pressure at 14th floor) at different height (relative to the altitude of 14th floor) was shown. It can be seen the calculated pressure exhibited a good linearity with the altitude in a small height range. From Fig. S9, by applying the relationship between $\Delta G/G_0$ and pressure that given in Fig. 2a, $\Delta G/G_0$ corresponding to each floor could be calculated. The results were plotted in Fig. 4c, together with the measured $\Delta G/G_0$ versus height curves. In the calculation, we assumed the height of each floor was 3.5 m. Please note, for convenience the conductance measured at 14th floor was used as G_0 in Fig. 4c. In the barometric altimeter experiment, the barometric pressure at 14th floor was about 500 Pa higher than the pressure of the reference cavity. This means there was a strain existed in the PET membrane when measuring G_0 in Fig. 4c, or say G_0 used in Fig. 4c corresponds to the conductance at 500 Pa pressure in Fig. 2a. Therefore, in the above calculation, when $\Delta G/G_0$ was read from Fig. 2a according to the calculated pressure, a 500 Pa shift on the pressure had to be considered. Fortunately, the abrupt slope change around 60 Pa in Fig. 2a was skipped, which more or less reduced the influence from pressure fluctuations and made the curves in Fig. 4c smooth.

The calculated $\Delta G/G_0$ verse height curve was noted in the revised manuscript (line 351, page 15):

Conductance response $\Delta G/G_0$ at different altitude was also calculated according to the $\Delta G/G_0$ versus pressure relationship shown in Fig. 2a and the barometric pressure calculated from a polytropic atmosphere model (see Supplementary 2.2). As shown in Fig. 4c, there is a good agreement between the experimental $\Delta G/G_0$ and the calculated one, verifying that the altitude measurement is reliable. The excellent sensitivity and resolution demonstrated a good operation of the device in practical situations remarkably. As a precise barometric pressure sensor, the sensor described in this paper could be applied as a high-resolution barometric

altimeter; such instruments are now widely required in vehicles, aircraft and personal navigation.

6. A curiosity. How does the hysteresis in Fig. 2(c) look for higher pressure? For example close to, and beyond, 50 KPa where the deformation may become plastic?

Answer: It is not sure the fluctuation on the conductance recorded along the pressure loading sequence in Fig. 2c in the original manuscript is related to the hysteresis. The fluctuation on the data may also come from the influence of various environment factors, such as vibration and air flow. To quantitatively study the hysteresis of the device, we increased the pressure applied to a sensor with a 0.05 mm-thick PET membrane up to 1 kPa and then released the pressure down to zero stepwisely, with the conductance corresponding to each step being recorded. The results were shown in Fig. S11a. It could be seen the hysteresis induced by a 1 kPa applied pressure was not significant. The relative conductance showed a shift of 0.012% from the initial value after the pressure loading-releasing cycle. This shift was equivalent to a conductance deviation induced by an applied pressure of 0.9 Pa, which is comparable to the resolution of our sensors. However, if similar pressure loading cycle was applied up to 100 kPa, the device exhibited a significant hysteresis, as shown in Fig. S11b. In this case, the PET membrane underwent a plastic deformation, which induced an unrecoverable change on the nanoparticle arrays. The large shift of the zero point made the device fail to work. To expand the working pressure range of the device, either a thicker membrane or a membrane with larger elastic modulus could be considered.

Fig. S11 Relative conductance changes of a sensor with a 0.05 mm-thick PET membrane subjecting to stepwisely changed pressure loading-releasing cycles within the pressure ranges of (a) 0-1 kPa and (b) 0-100 kPa.

Some hysteresis test data were added in supplementary information 3.2: (line 187, page 14)

3.2 Hysteresis in sensors

We studied the hysteresis of the device with a 0.05 mm-thick PET membrane. We increased the pressure applied to the sensor up to 1 kPa and then released the pressure down to zero stepwisely, meanwhile the conductance corresponding to each step was recorded. The results were shown in Fig. S11a. It could be seen the hysteresis induced by a 1 kPa applied pressure was not significant. The relative conductance showed a shift of 0.012% from the initial value after the pressure loading-releasing cycle. This shift was equivalent to a conductance deviation induced by an applied pressure of 0.9 Pa, which is comparable to the resolution of the sensors. In Fig. S11b, the largest applied pressure in the hysteresis test was increase to 100 kPa. The PET membrane underwent a plastic deformation, which induced a significant hysteresis.

A statement has been added in the revised manuscript (line 313, page 13):

...As shown in ~~Figure~~ Fig. 3a, the pressure sensor with a 0.05 mm PET membrane can work normally at 0–1.0 kPa differential pressure regime with an insignificant hysteresis (see Fig. S11a). Beyond this pressure, the sensor could still response well to the differential pressure up to 40.0kPa, with a sensitivity $S = -0.0037 \text{ kPa}^{-1}$ (negative sensitivity indicates that the tensile strain dominates the deformation on the PET membrane). But at ~~a~~ higher pressure, the deformation of the membrane may be plastic so that significant hysteresis emerges (see Fig. S11b) the monotonic response is lost, though there is still a good linear response in the pressure range 1.0–40.0 kPa, with a sensitivity $S = -0.0037 \text{ kPa}^{-1}$ (negative sensitivity indicates that the tensile strain dominates the deformation on the PET membrane).

7. It seems the actual invention is the device with array on PET polymer. For example see, Segev-Bar, Tunable Touch Sensor and Combined Sensing Platform: Toward Nanoparticle-based Electronic Skin, DOI: 10.1021/am400757q. Is there any special reason for good particle/polymer adhesion? Why not use Au nanoparticles?

Answer: Generally, NPs in the cluster beam generated from a gas aggregation cluster source could gain high flight speed when they pass through the gas dynamical nozzle. And they could impact on the substrate surface with a kinetic energy up to ~1 keV (see K Wegner *et al.*, Cluster

beam deposition: a tool for nanoscale science and technology, *J. Phys. D: Appl. Phys.* 39 (2006) R439–R459). Such an impact energy is high enough to displace at least one surface atom on the substrate, thus creating a reactive site that pins the NP to the surface, especially in the case of polymer materials (see R. E. Palmer K Wegner *et al.*, Nanostructured surfaces from size-selected clusters, *Nature Materials* volume 2, pages 443–448 (2003)). As a result, a good particle/polymer adhesion can be reached.

Some brief descriptions were added in the revised Supplementary information 1.1: (line 11, page 1)

Palladium (Pd) NPs were generated from a magnetron plasma gas aggregation cluster source in argon stream at a pressure of about 80 Pa and extracted to a high vacuum deposition chamber with a differential pumping system.^{S1,2} **And they could impact on the substrate surface with a kinetic energy up to ~1 keV.^{S3} Such an impact energy is high enough to create a reactive site that pins the NP to the polymer surface to form a good particle/polymer adhesion.^{S4}**

Generally, noble metals such as gold and silver are preferentially considered in many conductance-based device applications owing to their excellent conductivity and chemical stability. To constitute percolative conducting NP arrays, we excluded Au and Ag since we found Au and Ag NPs had high mobility and were easy to coalesce at many substrate surfaces, leading them to grow into larger particles rather than forming dense arrays at high NP deposition mass, as shown in Fig. S4a and b in the revised Supplementary. This behavior might induce significant performance degradation as well as unreliable and unrepeatable measurement results when Au or Ag NPs were used to constitute percolative NP arrays for piezoresistive sensing.

On the contrary, in the case of Pd nanoparticles, little coalescence among particles could be observed even at a very high NP density, so that perfect percolative NP arrays could be formed at moderate deposition mass (see Fig. S4c). This behavior is important to achieve a high stability in sensor applications.

In the revised Supplementary 1.2, TEM images of Au, Ag and Pd NP arrays were added, from which the NP aggregation status we described above was clearly demonstrated: (line 66, page 4)

NP arrays of Au, Ag and Pd with different coverages were prepared by cluster beam deposition.

The aggregation status of the NPs in the arrays was characterized via TEM (Fig. S4). Significant spontaneous coalescence and growth among NPs could be clearly observed in the Au and Ag NP arrays. This irreversible growth behavior will greatly influence the distribution of the nanoscale gaps in the NP arrays and induce large instability on the measured intrinsic conductance of the NP arrays. On the other hand, evidence of coalescence and growth among NPs was hard to be observed in the Pd NP arrays. Nanoscale gaps could be clearly observed between almost all the adjacent NPs. Obviously, Pd NPs were more suitable to be used to constitute percolative conducting NP arrays.

Fig. S4 TEM images of (a) Au, (b) Ag and (c) Pd NP arrays with (I) lower and (II) higher coverages. The scale bars in all images are 100 nm.

A statement has been given in the revised manuscript (line 147, page 7):

Generally, NPs of various metals can be used as the piezoresistive sensing medium. In the present research, Pd NPs were used in preference owing to their less coalescence (Supplementary Fig. S4) and chemical stability. Although gold has been preferentially

considered as a conductive and chemically stable element in many sensing applications,^{34, 35}
the high mobility and easy coalescence behavior of gold NPs leads to large instability when
they were used to constitute percolative conducting NP arrays.

8. Incomplete literature. The idea of modulating the conductance of nanoparticle array by deforming the underlining membrane is shown before to sensitively measure humidity and pressure variation due to sound (see Berry et al., *Angewandte Chemi Int. Ed.* 2005, 44, 6668; *Nano Letters* 2004, 4, 939). *Angewandte Chemi Int.Ed.*2005, 44, 6668; *Nano Letters* 2004, 4, 939)

Answer: We cited the two contributions and the literature in question 7 in the revised manuscript (line 51, page 3):

Recently, percolative nanoparticle (NP) arrays have been used as piezoresistive transducers of ultrasensitive mechanical sensors, such as strain sensors,^{17, 18} humidity sensors,¹⁹ as well as force and mass sensors.²⁰

And (line 58, page 3):

As a result, the conductance of percolative NPnanoparticle arrays is sensitively related to the deformation of the substrates on where the NPsnanoparticles deposite.^{17, 18, 20, 23}

And (line 293, page 12):

A report has shown that modifying the mechanical and geometrical properties of the flexible substrates could change the measuring range of sensors.⁴⁹

19. Berry V, Saraf RF. Self-assembly of nanoparticles on live bacterium: an avenue to fabricate electronic devices. *Angew Chem Int Ed Engl* **44**, 6668-6673 (2005).
23. Berry V, Rangaswamy S, Saraf RF. Highly selective, electrically conductive monolayer of nanoparticles on live bacteria. *Nano Lett* 4, 939-942 (2004).
49. Segev-Bar M, Landman A, Nir-Shapira M, Shuster G, Haick H. Tunable Touch Sensor and Combined Sensing Platform: Toward Nanoparticle-based Electronic Skin. *ACS Appl Mater Interfaces* **5**, 5531-5541 (2013).

9. Minor issues: (a) Personally, I find the description "quantum conductance" somewhat odd

terminology. (Generally, as the authors well knows, conductance is quantum mechanical). May be think of a better descriptor. (b) The term "pressure-dependent deformation" on line 132 does not mean anything to me. A more detailed explanation of the abrupt drop in sensitivity and yet maintaining linearity in Fig. 2(a) would be helpful.

Answer: According to the reviewer's suggestions, the using of "quantum conductance" has been replaced with "tunneling conductance" or "conductance of the array" in the revised manuscript.

In the original manuscript, "pressure-dependent deformation" was used to express that the pressure could change the deformation features of the PET membrane. It's really somewhat meaningless so that now it has been replaced with "transition of the pressure-dependent deformation". (line 184, page 8)

As mentioned in the response to comment 1, the $\Delta G/G_0-\Delta P$ curve exhibits approximate linearity in the low-pressure range, but tends to deviate from linearity at higher ΔP . An abrupt drop in sensitivity occurs at $\Delta P \sim 60\text{Pa}$. This characteristic is related to the developing of tension-strains that replace compress-strains on the surface of PET membrane with the increase of the pressure, as illustrated by the FEA results. These points have been explained in detail as response to comment 1, 3 and 4 of Reviewer #2. (line 204, page 9):

The inhomogeneous deformation of the membrane under pressure induces an inhomogeneous distribution of the inter-particle distance changes in the NP arrays. The conductance measured across the electrodes is an integration of the electron transport over all the conductive percolation pathways which contain various inter-particle distances characterized with a complex function of position and pressure. Therefore, the response of conductance to pressure is no more a simple exponential function of pressure. At a smaller applied pressure, the compressive strain dominates the main area of the membrane, so that the whole NP array undergoes a conductance enhancement. An approximately linear dependence between conductance and pressure is observed. With increasing applied pressure, a transition from compressive to tensile strain can be observed as the position changes from center to edge of the membrane (see Fig. 2c). The extension of the tensile strain regime makes part of the NP array undergoing a conductance reduction, which diminishes the increase in conductance generated in the central area. A drop on the conductance response curve can thus be seen.

Reviewer #3

Thanks for the holistic review and the suggestions for improving the quality of the manuscript. Here, we have added some discussions about the physical mechanism, which we hope will be acceptable to the reviewers.

1. The authors claim that quantum conductance is the mechanism responsible for the extremely high sensitivity of their device, however the physical aspects of this claim are not discussed. In particular the use of Pd particles should be taken into account from this point of view since Pd nanoparticles are prone to oxidation in a very wide range of conditions (J. Phys. Chem. C 2007, 111, 2, 938-949). Considering the production conditions used by the authors, one should expect to have oxidised Pd nanoparticles and PdO_x is a p-type semiconductor with a relatively low work function. These aspects should be discussed in order to provide convincing motivations about the conduction mechanisms.

Answer: Oxidation of metal nanoparticles under ambient condition is a common concern in many cases. It indeed has a significant influence on the conductance of the closed spaced nanoparticle arrays. For example, an oxide layer is formed on the Al nanoparticle surface in a short time after it is exposed to air and saturated at about 2.5 nm (V. I. Levitas, J. McCollum, and M. Pantoya, Sci. Rep. 5 (2015) 7879; A. L. Ramaswamy, and P. Kaste, J. Energ. Mater. 23 (2005) 1-25.). As a result, the conductance of the Al nanoparticle arrays decreases continuously and ultimately becomes too small to be measurable. For Pd nanoparticles generated with a gas phase cluster source, the existence of PdO_x layers on the nanoparticle surfaces is confirmed by conducting XPS and HR-TEM characterizations. However, we found the oxide layer should be no thicker than 0.5 nm, even if the Pd nanoparticles were exposed to air for a month. Meanwhile, the conductance of the Pd nanoparticle arrays remained stable. Considering PdO_x is a p-type semiconductor with a relatively low work function, the conductance of the nanoparticle arrays is possible to contain the contribution from the electrons emitted from the nanoparticle surface under the applied bias voltage. However, since more than one inter-particle nanoscale gaps may be contained in most of the percolation paths and the applied bias voltage in the measurement is only 1V, the voltage drops on each nanoscale gap should generally not be enough to stimulate the electron emission so that the conductance

measured from the nanoparticle arrays is not dominated by the electron emission mechanism. Therefore, we believe that the effect of the extremely thin PdO_x surface layer on electron transport is mainly a modification on the resistance of the tunneling junctions by the reduction of the inter-particle tunneling barrier height. The conductance of the NP array and the response characteristics of the device will be stabilized to a new value after the exposure to air.

The characterization of the oxide state of the Pd NPs were shown in the revised Supplementary 1.2: (line 80, page 5)

Fig. S5a shows the XPS (X-ray photoelectron spectroscopy) spectrum of Pd NPs. The nanoparticles were deposited on a quartz substrate with a deposition rate of 0.2 Å/s for 10 minutes, and placed in air environment for 30 days. The peaks of Pd 3d 5/2 and 3d 3/2 core levels are situated at the accepted binding energies for metallic Pd but contain broadened tails at the higher energy sides, which can be decomposed to additional smaller peaks corresponding to the core levels of Pd oxide, indicating that the Pd NPs are partially oxidized. The higher energy tails were eliminated after the sample surface was cleaned with Ar ion sputtering, indicating that the oxidation remained on the nanoparticle surface.

In Fig. S5b, HR-TEM image of a Pd nanoparticle is shown. Lattice images can be distinguished in the core, implying it is metallic Pd. Meanwhile, a thin amorphous shell, which is most likely PdO_x, is also distinguishable. Its thickness is measured to be about 0.5 nm on average.

Fig. S5 XPS and HR-TEM characterizations of the Pd NPs aged for 30 days. (a) The XPS spectrums of Pd NPs array in the regime from 320 to 350 eV. The accelerated argon ion flow was used to etch off the surface about 3 nm thickness. (b) A typical HR-TEM image of Pd NPs. An extremely thin amorphous oxide

layer on the surface of the NP was observed, and its thickness is about 0.55 nm.

A discussion about the conduction mechanisms of the Pd nanoparticle arrays with the consideration of the PdO_x surface layers has been added in the revised manuscript: (line 153, page 7):

Oxidation of metal nanoparticles is a common concern for a device working under atmospheric ambient directly. Oxidation of Pd NPs have been observed in a wide range of conditions.³⁴ For the gas phase deposited Pd NPs, our X-ray photoelectron spectroscopy (XPS) and High resolution transmission electron microscopy (HR-TEM) characterizations demonstrated that there is a PdO_x layer of 0.5 nm in thickness formed on the NP surface (Fig. S5). Considering PdO_x is a p-type semiconductor with a relatively low work function, the conductance of the NP arrays is possible to contain the contribution from the electrons emitted from the NP surface under the applied bias voltage. However, since more than one inter-particle nanoscale gaps may be contained in most of the percolation paths and the applied bias voltage in the measurement is only 1V, the voltage drops on each nanoscale gap should generally not be enough to stimulate the electron emission so that the conductance measured from the NP arrays is not dominated by the electron emission mechanism. Although the multi-barrier tunneling nature of electron transport in the NP array remains unchanged, the conductance of the array and therefore the response characteristic of the device will be stabilized to a new value after its exposure to air, due to the reduction of the inter-particle tunneling barrier height resulting from the formation of the PdO_x surface layer with a relatively low work function.

2. Another very important aspect is the percolation curve reported in the supplementary information section (Fig. S2). The shape of this curve should depend on the nanoparticle deposition conditions: the control and reproducibility of these parameters and of the percolation curve should be discussed in the main text of the manuscript, since this is a very important aspect related to the possibility of producing reliable and reproducible devices. A critical issue is the change of slope of the curve reported in Fig. S2 from which the authors determine when to stop the film growth. The authors should discuss in detail the physical reasons of this sudden change in slope since these are connected with the nature of the electrical behaviour of the sensor and with the mechanism (quantum conductance?).

Answer: As suggested by the reviewer, we moved Fig. S2 to the main text of the revised manuscript as Fig. 1d. Some schematics were added to help to understand this curve. We also added a new figure (Fig. S2) to demonstrate the possibility to control the conductance evolution (the percolation curve) shown in Fig. 1d with the NP deposition conditions. We showed that the shape of the curve could be well adjusted by controlling the NP deposition rate. Considering that a stable deposition condition can be finely maintained in the gas phase cluster beam deposition, together with a real time conductance monitoring, the reliability and reproducibility of the device fabrication could be assured. About the physical reasons of the slope change of the conductance evolution curve accompanying with the NP deposition, we have added a detailed discussion in the revised manuscript based on the quantum tunneling nature of the conductance of the NP arrays, by taking into account the specific electrical configuration of the devices. We stated that (line 104, page 5):

The conductance of the NP film was monitored during the deposition process (Fig. S1c). A typical conductance evolution curve measured during the NP deposition process is shown in Fig. 1d. It exhibits the characteristics describable with a percolation model,^{30, 31} considering the similarity between the coverage of the NP assembly and the particle filling fraction used in the percolation model, both of which increase with the deposition time. In our device, the electrodes cover an area of several to several ten square millimeters, resulting in a huge aspect ratio (total electrode length vs. electrode separation) of the inter-electrode gaps. This morphology leads to a rapid increase of the conductance after the NP coverage reaches the percolation threshold, which is determined by the electrode separation, due to the formation of a large number of conductive percolation pathways (or say closely spaced NP chains across the electrodes). Furthermore, due to the quantum tunneling nature of electron transport, the development of the conductance during NP deposition is not only dependent on the geometric filling pattern of the NPs but also dependent on the distribution of inter-particle gaps along the conductive percolation pathway, which also changes with the increase of the deposition mass. As a result, a finely gradual change on the slope of the conductance evolution curve reported in Fig. 1d can be observed in the vicinity of the percolation threshold after a few number of

conductive percolation pathways are formed, which enabling a least measurable electric current. With a further increase on the deposition time, the conductance evolution curve soon reaches a rapid rising slope and the gradual change of the conductance with such rising slope can span four order of magnitude or more, rather than displaying an uncontrollable sudden drastic rising as many classical granular conductive composites show. This nature is important for the device to achieve a sensitive and high resolution response to the change on the inter-particle gap distribution induced by various physical actions, such as pressure and strain. The slope of the conductance evolution curve can be finely adjusted by the NP deposition rate, as shown in Fig. S2. Since a precisely controlled deposition rate can be maintained in the gas phase cluster beam deposition, a reliable and reproducible production of the devices can be assured.

Fig. 1 has been modified:

Figure-Fig. 1: A **barometric**-pressure sensor based on the Pd NP arrays and its fabrication and

characterization. (a) A schematic diagram of the piezoresistive barometric pressure sensor prototype in a quarter cross-sectional view. (b) ~~The operating principle of our sensors. These red lines linking NPs represent percolation pathways which may exist in arrays. When a pressure is loaded, more percolative pathways are generated in the compressed PET membrane, making the conductance of NP arrays increase from G_0 to G_1 as the inner shown.~~ (c) Photograph of the actuation layer of the sensor, consisting of a PET membrane covered with Pd NP arrays and Ag IDEs. (ed) ~~Evolution of the conductance between the IDEs during NP deposition. The bias voltage was 1 V. Deposition was stopped at $G = 250$ nS. The insets show a schematic depiction of the inter-particle electron tunneling pathways in metal NP arrays before, close to and beyond the percolation threshold.~~ ~~The operating principle of our sensors. These red lines linking NPs represent percolation pathways which may exist in arrays.~~ (de) HAADF-STEM image of the Pd NP arrays. (ef) Size distribution of the NPs measured from (de).

To demonstrate the possibility of the controlling of the conductance evolution during NP deposition so as to realize a reliable and reproducible production of the devices, we added two additional conductance evolution curves measured at different NP deposition conditions in the revised Supplementary (Table S1 and Fig. S2). It confirmed that the rising slope of the conductance evolution curve shown in Fig. 1d (in the revised manuscript) is positively correlated with the deposition rate of the metal NPs. The detail is as follows in Supplementary 1.1: (line 26, page 2)

A typical plot of the conductance evolution of the NP arrays, which can be described by using the percolative **electron transport** model, as a function of deposition time is shown in ~~Figure S2~~ Fig. 1d. ~~Currents across the IDEs gap are almost zero in the initial stage of deposition. As more NPs are deposited onto the surface, the inter-particle distance reduces and current pathways gradually form throughout the IDEs. As a result, the current begins to increase abruptly at 400 s corresponding to the percolation threshold. The deposition was stopped when the predetermined conductance values (~ 100 nS in this work) were attained. Finally, a sensing element based on Pd NP array with predetermined conductance value was obtained. After approaching the percolation threshold, a continuous increasing of the conductance with the deposition time, including a fine change on the rising slope of the curve near the least measurable conductance was observed. Here we demonstrate that the rising slope is positively correlated with the deposition rate of the metal NPs. Pd NP depositions with rates of ~ 0.2 , 0.25 and $0.3 \text{ \AA} \cdot \text{s}^{-1}$ were performed by controlling the discharging power at 27, 32, and 37 W in the~~

cluster source, respectively. The main operation parameters for depositing NPs were summarized in Table S1. Beyond the percolation threshold, the conductance of the NPs array increased with the deposition time with a gradually changed rising slope. A fine control on the rising slope with the deposition rate was shown (Fig. S2). The higher the deposition rate, the quicker the conductance increasing, resulting in a steeper rising slope.

Table S1 Main operating parameters for Pd NPs deposition

Parameters	Values
Aggregation tube pressure (Pa)	80
Discharging power (W)	27, 32, and 37
Deposition rate (Å/m)	0.20, 0.25, and 0.30

Fig. S2 Evolution of the NP arrays conductance at different deposition rates. t and $t_{\text{threshold}}$ denote the deposition time and the time when the percolation threshold is reached, respectively.

REVIEWERS' COMMENTS:

Reviewer #2 (Remarks to the Author):

I find the response satisfactory.
The authors have provided an exhaustive response.
I leave it to the editors regarding the length of the MS.

In my opinion the MS is acceptable for this fine journal, Nature Comm.

Reviewer #3 (Remarks to the Author):

The authors have addressed in a convincing and exhaustive way the issues raised by the reviewers

Reviewer #2 (Remarks to the Author):

I find the response satisfactory.

The authors have provided an exhaustive response.

I leave it to the editors regarding the length of the MS.

In my opinion the MS is acceptable for this fine journal, Nature Comm.

Answer: The authors appreciate the referee for the support and approval of our manuscript sincerely. In order to comply the police requirement on manuscript length of *Nature Communications*, we have finally revised our manuscript. Now, the number of words in the main text remains less than 5000, meanwhile the integrity of the manuscript content is maintained. Finally, the authors wish to express sincere thanks to you for the comprehensive suggestions and comments which could improve the quality of our manuscript.

Reviewer #3 (Remarks to the Author):

The authors have addressed in a convincing and exhaustive way the issues raised by the reviewers

Answer: The authors express their sincere gratitude to the referee for that the discussion on the physical mechanism suggested by the referee could be approved. Thank the referee for all of these comments and suggestions during the reviewing.